# Period poverty: The perceptions and experiences of impoverished women living in an inner-city area of Northwest England

**Madeleine Boyers[1,2], Supriya Garikipati [3], Alice Biggane[1], Elizabeth Douglas[4], Nicola Hawkes[5], Ciara Kiely[1], Cheryl Giddings[1], Julie Kelly[6], Diane Exley[7], Penelope A. Phillips-Howard[1], Linda Mason [1] ***

**1** Department of Clinical Sciences, Liverpool School of Tropical Medicine, Liverpool, England, **2** Liverpool Women's Hospital, Liverpool, England, **3** University of Liverpool Management School, Liverpool, England, **4** The Whitechapel Centre, Liverpool, England, **5** South Liverpool Foodbank, Liverpool, England, **6** Adelaide House, Liverpool, England, **7** Brownlow Health, Liverpool, England

* Linda.Mason@lstmed.ac.uk

## Abstract

### Background

The menstrual needs of girls and women are important to health, education, and well-being. Unmet need and harm from poor menstrual health in low-and- middle-income countries have been documented, but with little empirical research undertaken in high income countries. Continuing austerity in the UK suggests menstruators are likely more vulnerable to 'period poverty' than previously, with the COVID-19 pandemic assumed to exacerbate the situation.

### Aim

To explore the menstrual experiences and perceptions of women in the UK who are living under circumstances of deprivation, alongside views of staff working in organisations supporting these women, to understand whether women's menstrual needs are met.

### Methods

A qualitative study was conducted in an inner-city in NW England. Three focus group discussions and 14 in-depth interviews were conducted across three study sites supporting impoverished women. Data was analysed thematically.

### Results

Themes were: reflections on menstruation; affordability of products; access to public facilities; organisational support; potential solutions. Many women perceived menstruation as a burden in three aspects: physical discomfort and pain; psychological anxiety; and shame and stigma. Managing menstruation was difficult due to cost relative to low incomes, with food, heating and lighting prioritised, leaving women improvising with materials or wearing products for longer than desired. Most suggested that products should be free, often

**Data Availability Statement:** All data are available from the UK Data Service Reshare [https://reshare.ukdataservice.ac.uk/855592/]. However the data have been placed under permission only access

due to the confidential nature of the data and the consent provided. For access, please contact Tracy.Seddon@lstmed.ac.uk or the corresponding author.

**Funding:** Preparation of this paper was supported by the UKRI 'Enhancing place-based partnerships in public engagement' grant. The reference number for this grant is UKRI/EPPE19/13 The grant holder was PPH. The funders of this study had no role in study design, data collection and analysis, decision to publish, or preparation of the manuscript.

**Competing interests:** The authors have declared that no competing interests exist.

remarking if men required similar items this would happen. Most women were unaware supporting organisations provided free products. Staff felt the small range of products offered did not meet client needs and were ill-prepared to have conversations on products and clients' menstrual needs.

## Conclusion

Impoverished women lack the necessary resources to manage their menses well which negatively impacts their health and brings stress, embarrassment, and shame. Support, including access to free products, is needed at both local and national level to help impoverished women manage their menstrual hygiene.

## Introduction

The menstrual needs of girls and women are increasingly recognized as an important issue for their health, education, employment, and well-being [1]. Much of the evidence-base to date has demonstrated unmet need and harm from poor menstrual health and hygiene (MHH) among girls in low-and-middle income countries (LMIC) [2] The term 'menstrual health and hygiene' has been described as the needs experienced by people who menstruate, including having safe and easy access to the information, supplies, and infrastructure needed to manage their periods with dignity and comfort (menstrual hygiene management) as well as the systemic factors that link menstruation with health, gender equality, empowerment, and beyond [1]. An inadequate supply of water, hygiene, and sanitation (WASH) facilities, a lack of information about menstruation and its management, and a lack of menstrual sanitary products affect the ability to manage menstruation hygienically, comfortably, and safely, impacting on psychosocial well-being, and sexual and reproductive health. Menstrual stigmas and taboos create further challenges [3]. In our paper we interweave the terms 'women' 'girls' and 'menstruators' as appropriate and according to sources of information we cite.

Until recently, efforts to improve MHH have largely focused on the needs of schoolgirls, and to a lesser extent women, in LMIC, with the assumption that the challenges for girls and women in high-income countries (HIC) are fewer and perhaps less significant [2]. However, increasing divisions in wealth disparity among populations in HIC have highlighted possible struggles for some sections of society to maintain basic levels of MHH [4]. The term 'period poverty' has been widely adopted, denoting an individual's need for menstrual products without adequate funds to purchase them [4]. Relying largely on anecdotal evidence to provide impetus, media campaigns have been established to generate awareness of period poverty to improve access to affordable menstrual products for all menstruators as a basic human right. However, stigma and societies' attitudes also need to change in parallel, to educate, support and empower menstruators, reducing period shame [5]. Whilst the present study focuses on period poverty, it is recognized this is just one facet of MHH that needs to be addressed.

There has been some momentum to make menstrual products more accessible including efforts in HIC. Canada and Australia removed the sales tax on menstrual products in 2015 and 2018 respectively, with the UK joining in 2021. In 2020, Scotland became the first country to make period products free for all which meant local authorities became legally obliged to provide free menstrual items to anyone that needs them [6]. In the rest of the UK, advocacy campaigns and actions (e.g. Pink Protest, Red Box presentations to Parliament) have resulted in legislation to provide free menstrual products in UK primary and secondary schools.

However, despite some success in reducing menstrual inequalities in, for example, the UK, Canada and Australia, there remains limited empirical evidence on period poverty and how to address it in HIC. A recent systematic review highlighted that most research on menstruation in HIC was designed to "understand the constructed meanings of menstruation" rather than informing practice or policy. Just six qualitative studies were identified that targeted participants of low-income status, with the authors concluding that to date there is insufficient evidence on the needs of marginalized and impoverished women [4]. The few studies conducted, mostly with a US focus, suggest a substantial proportion of respondents reports period poverty on occasion, with a core group unable to purchase sufficient menstrual products every month. A quantitative study among US women on low incomes found 64% reported period poverty during the previous year, a monthly occurrence for one fifth of respondents [7]. Another study among American women found that 81% of women receiving treatment for substance abuse reported 'menstrual poverty' [8]. Both studies found a significant association between period poverty and food insecurity. Period poverty was also documented in a small US study of 58 high school students [9] with almost half of the students unable to afford products during at least one menses in the school year, whilst 12% were unable to afford them 'most' months [9]. A recent study found one quarter of 471 US college students reported period poverty in the past year, including 10% for whom it was a monthly occurrence [10]. A phenomenological study interviewed 40 'vulnerable' women in the UK (i.e. women accessing homeless shelters, drug support groups, day centers, and foodbanks). Most described their experience of menstruation in a negative manner and disclosed a need for privacy in self-care, as well as access to affordable menstrual products [11]. Having to ask friends or staff for products in institutional settings was a common but shameful experience. Many resorted to makeshift absorbents such as toilet tissue; others resorted to shoplifting to meet their needs. Two qualitative studies among homeless women in the US similarly reported that women were reduced to shoplifting for supplies or going without food in order to purchase them [12,13]. Similarly, Kuhlman and colleagues, also in the US, found many low-income participants relied on donations at times and reported using cloth, rags, tissue paper, or paper towels as absorbents when necessary [7]. Some reported they had to choose between purchasing food or appropriate menstrual products. Similar experiences were documented in the report by 'No More Taboo', a not for profit enterprise, who interviewed 37 women from low-income backgrounds in the UK, and found that half the sample reported menstrual products as too expensive, with a quarter having to improvise with t-shirts, toilet roll and nappies [14]. Women also reported suffering physical symptoms, stigma and embarrassment. Other UK studies have noted use of child nappies and incontinence sheets to deal with heavy bleeding [15,16].

The current COVID-19 pandemic has 'amplified' the marginalisation and vulnerability of menstruators worldwide, according to Crawford and Waldman [17]. They identified four key areas of concern: access to affordable products; maintenance of hygiene and sanitation; menstruation information and support; and privacy and dignity issues arising through restricted movement.

Similarly, Plan International [18] conducted a small survey of global WASH professionals who identified that difficulties already ongoing were made even more challenging through restricted access to products, WASH facilities, information, and an increase in the price of the products. Such challenges appear not only confined to menstruators in LMIC, but also are experienced amongst those living in the UK and other HIC. Plan International's UK survey [19] conducted during the pandemic found 11% of girls aged 14–21 could not afford period products and were improvising with makeshift products, whilst 22% of girls who could afford period products were struggling to access them, mostly due to a lack of availability in shops. With continuing austerity and the subsequent increase in demand for foodbanks and homeless

shelters, menstruators in the UK and elsewhere appear more vulnerable to menstrual hygiene issues than ever before [20]. Thus, despite recent policy changes, formal research evaluating the scope and health risks associated with period poverty in the UK is required to determine the wider public health implications among the most impoverished populations, and to examine whether the recent policy changes go far enough.

The present study aimed to explore the menstrual experiences, needs and perceptions of women living in circumstances of deprivation, through engagement with women and with local partners assisting with their welfare needs. We also aimed to gather perspectives from partner organisation staff to help contextualize their understanding of women in need and assess any met or unmet support they provide, along with any other concerns they identified.

## Methods

### Study design

This was a qualitative study using focus group discussions (FGD) and in-depth interviews (IDI) among women living in circumstances of deprivation, and staff volunteering or working in services assisting such women.

### Study setting

The study was set in a city in northwest England, ranked within the top five most deprived areas in England. Three organisations were purposively chosen to represent women living in impoverished circumstances, along with staff who work within the organisations. Site 1 is a hostel for homeless women; site 2 falls under 'Approved Premises' which provides care and services for women after incarceration; and site 3 is a foodbank within the inner-city which caters for people on low-incomes, including those who become unemployed. Site 3 includes women who may have more settled accommodation. Sites 1 and 2 are referred to as 'supported accommodation sites' throughout the study.

### Study tools

The content of the FGD and IDI guides was informed by a literature and media review and developed by the research team (See S1–S3 Files).

The FGD and IDI topic guides began with the question "Can you tell us what menstruation means to you?" which aimed to open the discussion around menstruation and give the researchers an idea of the participants' first thoughts on the subject. Further topics covered menstrual symptoms, the lead up to a period, access to menstrual products, and WASH. The topic guide ended by inviting opinions on any changes the interviewee would like to see. Topic areas for staff IDI included the nature of their service, the type and needs of service users, the provision of any menstrual health products or services, unmet menstrual needs and suggestions for any menstrual health improvements.

### Eligibility

Women were eligible to participate if they were 18 years or older, had reached menarche but not menopause, and had a recent menses (within the last three months). Staff were required to be 18 years or older. As they were not discussing their own experiences with menstruation but representing organizational perspectives and their experience as staff, there were no eligibility criteria pertaining to their own menses, and both female and male staff were equally eligible.

## Sampling

Purposive sampling was used for the FGD and convenience sampling was used to sample for the IDI.

## Participant recruitment

For the FGD, a member of staff from each site identified and approached potential participants on our behalf. They provided written information, answered any initial queries and checked eligibility. They suggested a mutually convenient time and date for the FGD which was held in a private room at that site. For IDI with women, researchers attended the Site 3 foodbank and recruited women opportunistically after handing out fliers outlining the project. They answered any questions that women had and arranged an interview time for those who were agreeable and met the inclusion criteria. Staff IDI were arranged through first discussion with staff in their organisation, with a follow-up by phone to arrange an interview time.

## Data collection

The research team collected data between February and March 2020. One of the authors (LM), provided training / re-training to all data collectors covering the principles of qualitative data collection, use of interview guides and ethical considerations. The IDI were conducted by one of five members of the research team (AB, CG, CK, PPH, SG); the FGD were moderated by LM with AB or CG acting as notetaker.

## Data analysis

All FGD and IDI were digitally recorded, with participant permissions, and transcribed verbatim by MB. Each transcript was read by the interviewer / moderator to check for accuracy and, in the case of the FGD, to provide any missing participant numbers where possible. Data were analysed by thematic analysis providing a systematic framework for analysing qualitative data [21]. A coding framework (see S4 File) was drafted separately by three members of the research team (MB, PPH, LM) following reading and re-reading of the transcripts to gain data familiarity. The researchers compared, discussed, and assimilated the coding frameworks into one. MB then coded all relevant data using the framework before organizing these into overarching themes and subthemes. This was verified and agreed (PPH and LM) before a comparison and contrast within and between the different groups of women was undertaken. A narrative was compiled and illustrated with quotes chosen to demonstrate best the point in hand.

## Ethical considerations

This study and research tools gained ethical approval from LSTM Research Ethics Committee (19–068) on 31st January 2020. The research team obtained informed consent from all participants prior to each IDI / FGD. All participants were provided with an information sheet and a consent form. As some of the women may have been illiterate, they were reminded verbally of the purpose of the study, what participation would involve and given the opportunity to ask further questions. They were asked to provide an 'x' to denote signature as a sign of consent. Staff were similarly reminded of the same information but provided their written signature to denote consent.

## Results

### Participant characteristics

A total of 32 women and 5 staff, including one man, participated in the study, 23 of whom contributed to FGD and 14 to IDI, with recruitment ending sooner than anticipated due to COVID-19 restrictions (Table 1). For quotes, an assigned code defines the IDI or FGD. Women's interviews were assigned 'W' before 'IDI' whilst staff interviews are denoted 'staff'. P [number] was used to represent the individual participants in an FGD with Px denoting missing participant number. M represents the FGD moderator, whilst I denotes the interviewer.

Five themes were derived from the data. These were: reflections on menstruation; affordability of menstrual products; access to public WASH facilities; organisational support; potential solutions.

**Reflections on menstruation.** Invariably, women's narratives (from both FGD and women's IDI) expressed negative perceptions of menstruation, suggesting that they experienced it first and foremost as a burden. The encumbrance comprised three key aspects: physical discomfort and pain (predominantly the first aspect they spoke of); psychological disturbance; and the stigma and shame associated with menstruation.

Many women spoke of their own trials with heavy menstruation, the associated pain and the impact this had on their lives, including needing time off work, being bedridden with pain, or generally being unable to function as usual during the onset or first few days of menses.

P *"but when I'm off the pill I get really bad pains and heavy periods"*

M *"ok and how long would they last, your periods?"*

P *"probably about 5 days"*

P *"and the period pain would be about 2 days. 2, 3 days. A day of that I'll be completely not able to do anything"*

P *"in bed with a hot water bottle or a bath or painkillers yeah"*

M *"yeah, so its"*

P *"yeah, completely debilitating"*

(W IDI03 Foodbank)

*"I know [menstruation] means pain and it means discomfort for a good few days and. . .it's just something us women have to endure unfortunately. It's something I hate about being a woman, having to go through a period every month. You know, I once passed out with the pain it was so bad, I had to be sent home from work."*

**Table 1. Participant affiliations.**

| Organisation | Interviews | | |
|---|---|---|---|
| | **FGD with women** | **IDIs with women** | **ID with staff** |
| Supported accommodation sites | 3 FGD with 23 women | 1 | 3 |
| Food bank | 0* | 8 | 2 |
| **Total no. of participants** | **23** | **9** | **5** |

*No FGD was possible at the foodbank because of the nature of the service (see under limitations).

(W IDI01 Supported accommodation).

Some participants described changing their daily routine to cope with their menstrual flow. This varied from staying at home to ensuring they spent limited time in public and were able to return home to change their menstrual protection.

*"I think I have quite heavy periods and a lot of the time I'll go out and I'll make sure I'm home in an hour or two cos if I have to change when I'm out it can be really messy because if you bleed really heavily and stuff, it restricts where I'm going and what I do when I'm bleeding"*

(FGD1 P3 Supported accommodation).

Psychological symptoms such as low mood, anger and worry before and during menstruation were described in all FGDs, and in over half of the women's interviews. These symptoms were often a concern; occasionally the emotions they evoked were felt to be out of the individual's control.

P1 *"I'm like when I'm due on, If I wasn't on anti-depressants, I have a really bad time"*

M *"yep"*

P1 *"emotionally. I'm crying at the drop of a hat and stuff like that but erm but being on anti-depressants has helped but, if I came off them, I would dread coming on every month."* (FDG02 Supported accommodation).

P4 *"it's daunting having your period, it's not a nice time of the month is it. It's just"*

M *"when you say it's daunting"*

P4 *"feel washed out and anxious and irritable and frustrated"*

P2 *"when you know it's happening you want it to just hurry up and happen"*

*(*FGD01 Supported accommodation).

Repeated throughout all FGD and many of the women's IDI, participants described the shame and anxiety arising from having a period, and stigma they felt by their menses. The word '*embarrassment*' was echoed throughout many of the IDI and FGDs. A key cause of worry was the fear of leaking in public, expressed frequently during the narratives, and appeared a pervasive cause of anxiety for many participants throughout their menses.

*"don't want it leaking into your pants and then it's on show. The embarrassment of walking round constantly checking yourself all the time"*

(FGD03 P7 Supported accommodation).

Just two participants from different FGD and 3 participants from women's IDI expressed positive aspects of menstruation, but only after specific questioning. When explored, thoughts focussed on menstruation as being a sign of health. For a couple of women, menstruation represented their fertility and being 'a woman', which was encouraging for them. Another participant associated menstruation with reassurance, as it was a sign that she was not pregnant.

*"for me, it's like I think that's my body, my healthiness coming back cos I don't have periods for like 3 years then I get like one out the blue then, sounds bad like but when I go to jail and I*

*get my body healthy again, it's the day I get out of jail I end up coming on my period, so for me it's like my body's way of being healthy again its coming back you know what I mean"* (FGD03 –P3 Supported accommodation).

We noted the participants making positive comments did not report physically painful or disruptive periods.

The overall participant view was that menstruation was a burden that women had to endure. The phrase, "*have to just get on with it*" was mentioned multiple times in different FGD and women's IDI.

**Affordability of menstrual products.**   Managing menstruation was difficult for many women because of the heavy financial cost relative to their incomes. Most of our participants were receiving Universal Credit from the government, a payment made to help with living costs for those out of work, unable to work or on low income. Women across both FGDs and IDIs reported the amount received was insufficient for daily living and spoke of how they struggled to manage their finances, particularly those supporting a family. Menstrual products were seen as 'extras'; consequently budgeting for them was difficult and often prioritised last over essential items such as food, lighting, and heating.

*"If it was the choice between five pound on (menstrual) products and five pound on gas and electricity, it's gonna be the gas and electricity."*

(W IDI03 Foodbank)

Women who had experienced living on the street described particularly hard struggles managing their periods under these circumstances. In addition to the cost of menstrual absorbents, women spoke of the supplementary financial burden of having to purchase extra detergents or soaps for washing or replacing soiled items of underwear or clothing. Further, to adequately manage their blood flow over the course of their period, many women talked of using their menstrual product for longer than they wished.

*"well I'll just give you an example . . . you know as I said we're struggling a little bit, and when I realized– 'cos me app it gives me, it tells me oh you're due on your period in a couple of days and I thought 'oh, god what have I got in?' you know what I mean because we've got no money at the minute. And, luckily, I did have about six [tampon brand] left so I thought you know what—and I know this sounds terrible—but with me being at home and that I'll just have to like you know budget the ones I've got and just make them last and if I've got to use tissue I'll use tissue in between."*

(W IDI05 Foodbank).

Although nearly all women reported that they had resorted to using toilet paper or tissue at some point to prevent leakage, often on their first day of menses (when unprepared for their period), for some this was a usual course of action when they were short of money. Indeed, improvisation of menstrual absorbent using items such as toilet roll, hand towel, pillowcase, socks, incontinence sheets, cotton wool or sponge was a common solution to not being able to afford products on occasion and described or agreed to (in FGDs) by most of the women.

*P1 "I've had to use socks before today"*

*P7 "I was just about to say, thank god you said that"*

*Px "I have done"*

*Px "I have done"*

*P1 "I'd rather, you know, use something clean than nothing at all, you know what I'm saying, I'll do what I've got to do, know what I'm saying"*

*P7 "don't want it leaking into your pants and then it's on show"*

(FGD03 Supported accommodation).

*"I've been so heavy and its usually fallen on a week where I don't have much money... and I have had to go into a shop and rob tampons to get by... Or sit on a towel, see me sitting on a towel and not able to go out the house"*

(W IDI01 Supported accommodation)

A few women, unable to budget for or afford menstrual products, reported having to borrow pads or tampons or even shoplift as a last resort. These women appeared embarrassed about their actions. One woman admitted to having engaged in sex work to purchase menstrual products.

*P7 "You can't afford to go and buy like 7 packets, can't afford to just go pay 3, 4 pound odd a packet do you know what I mean, that'd be most of my money, do you know what I mean, and I'll be honest, [tampon brand] I robbed cause they're smaller"*

*P1 "I robbed them, because I can't afford to pay for them"*

*F1 "how do you feel about having to do that?"*

*P7 "get caught, the embarrassment of getting caught and you're taking that risk of reoffending and going to jail "what are you in for?" "a box of tampons" (laughs)*

*F1 "do you think its common that people have to do that"*

*P1 "yeah"*

*P2 "yeah it will be. If they're honest, yeah, especially in this environment"*

*(FGD3 Supported accommodation)*

*"I work in a big warehouse thing right and got caught last month, like come on and was like shit, so like obviously we sell sanitary products, so like a damaged one, I like opened it, just so I could take a few and I thought if I get caught here now or something doing that, is it worth losing my job over, just pinching a few tampons"*

(FGD2 P4 Supported accommodation).

*I "You also mentioned that you did some sex work, to get money"*

*P "yeah, years ago"*

*I "erm did you ever exchange sex for money so you could buy menstrual products?"*

*P "yeah, yeah definitely"*

(W IDI01 Supported accommodation).

Participants who complained of heavy bleeding reported they had to use many products and therefore spent *'a fortune'* each month.

When asked, nearly half of our participants felt that menstrual products were *'well expensive'* although most resorted to purchasing shops' own-brand products when possible, going to discount shops rather than local corner shops or supermarkets to get their products at the cheapest price.

> *"I think the pound shop is great cos otherwise corner shops or even the big supermarkets, tampons they're sort of £2.50, or £3, so if you go to the pound shop they're only a pound."* (W IDI02 Foodbank).

Not all the women shared these views. A few participants thought that menstrual products were affordable, with one woman having the view that they are essential, so women must buy them regardless of the price. However, as essential items, many participants felt that they should be provided free, comparing them to condoms, and often remarking that if men required similar items, they would be made freely available.

> *"it's not a luxury to prevent having blood running down your legs is it. That's not a luxurious thing, that's just practicality"*

> (FGD2 Px Supported accommodation)

> *"I think that as well the way you think you have to go and buy sanitary towels, tampons and stuff like this yet you can go to the sexual health clinic and get condoms for free. Do you know what I mean, you chose to have sex, you don't choose to have periods do you know what I'm saying so I don't get why. . .why you have to pay for them"*

> (FGD3 P5 Supported accommodation).

> *"I think they'd be much cheaper if men had periods. . .probably free don't you think"*

> (W IDI01 Supported accommodation).

Women spoke only of commercially available sanitary pads or tampons and were therefore asked their thoughts about reusable products. Participants had little or inaccurate knowledge about the reusable products available. There were mixed opinions and reactions towards reusable products, with no organisation supplying these types of products. Just five women had heard of menstrual cups. Positive responses from these women included discussion on the lack of chemicals involved and the long-term cost effectiveness.

> *"I'm not personally disgusted by it and I think it's a good idea and . . .. tampons and that have bleach in them so if you're using tampons, you're inserting all those chemicals into your body. So, like menstrual cups are a good idea but it's the practicality of them, like public toilets, if you have to take them out and then wash it, most public toilets, you'd gotta go into the cubicle, empty it, come out of the cubicle, use the sink to wash it, again it's not very private because you're in the open area of the toilets"* (FGD 01 Px Supported accommodation)

Most participants, when informed by the researchers about menstrual cups, wanted more information on them to decide if they would consider using them. Some participants in the FGDs found the idea of a reusable cup "*disgusting*" with concerns raised about the smell associated with menstrual blood. Others were curious but put off by the initial cost. Washing in public was highlighted to be a likely issue, with public toilets not thought suitable places to change menstrual cups.

*"but if you're struggling to buy towels or tampons. I can't, the ones I've looked into was like £30 or something like that, that's a lot in one lump sum if you're struggling to pay for any other kind of sanitary products"*

(FGD02 P3 Supported accommodation)

Only one participant in the women's interviews knew of reusable sanitary pads. All IDI and FGD that discussed reusable sanitary pads had a negative response, finding the concept '*backward*', '*dirty*' and '*time-consuming*'. The increased washing involved and maintaining good hygiene when in public was a particular concern for these participants.

**Access to public WASH facilities.**   When asked about managing menstruation in public, the majority of the women's responses were negative. Topics discussed were the lack of access to and payment for toilet facilities, disposing of products, and keeping clean during menses. Having to pay to use public toilets was acknowledged as a problem, and whilst some participants were seemingly more accepting of the situation, others were aggrieved by what they felt was an unfair situation, further compounded by the lack of available facilities. Women noted there used to be more public toilet facilities in the community, but these are now few or non-existent in some areas.

*Px "yeah you have to pay like 20p or 50p or something erm for the toilets with the automated doors and all that".*

*P6 "yeah the ones in the train stations they are about 40p or something now aren't they. Have to pay to go to the toilet, it's ridiculous!"*

*P2 "yeah that's ridiculous, especially for us women, you know you're not really having the choice are you, if you've gotta go you've gotta go"*

*(*FGD 02 Supported accommodation)

Some women spoke of finding restaurants with free facilities to use, but described difficulties, as many are for customer use only. This resulted in their having to purchase something (which they could ill-afford) or being refused entry, which was described as shameful.

*"They'd say 'Toilets not working' I just think they didn't want me in you know to go and change"*

(W IDI01 Supported accommodation).

Without prompt, a participant brought up the difficulties of menstruating whilst living on the streets and commented on the struggles with washing and accessing products with her living situation. Other participants in the FGD agreed and shared similar experiences. Mention was made of occasionally being told to leave the toilets of private food/drink chains on the assumption that they were accessing them to take drugs.

*"I think (mumbles) when they're in a longer time than usual, do you know what I mean, because say if I'm here I'd change them more often but if you're on the street and you don't have that access you do try and have them on longer and that's when they will start chafing"*

(FGD 02 P1 Supported accommodation)

*P1 'its horrible living on the street . . .it's hard enough to get washed and you can't in the shop and sometimes I've had to go into the shop and rob them [products], being honest . . . they've been that bad'*

*Px 'you going in toilets in likes of [fast food chain]'*

*P2 'and you're using tissue and wipes, baby wipes, to wipe yourself down below . . .because you haven't got access to somewhere to have a shower or a bath'*

*P1 'and it's not nice 'cos it's not even that, people know you are homeless in town so they don't even want you going into the toilet'*

(FGD 03 Supported accommodation)

Disposal of products did not seem to be an issue for participants however, as women reported using sanitary bins where possible, or they disposed with household waste in their homes.

**Organisational support.** Most of the women interviewed at the foodbank were first time users of the service and were not aware that this facility provided menstrual hygiene products. Similarly, most participants in one FGD comprising women accessing supported accommodation were surprised when the availability of menstrual products was mentioned. There was a lack of dialogue around how the women perceive the services to support their menstrual needs. None of the women described the organisations as having any part in their menstrual management.

On the other hand, the staff interviewed, spoke of their organisations providing free menstrual products, relying on donations to supply these items. The foodbank buys and restocks additional sanitary items if their donations are depleted, to ensure they have a continuous supply for service users. When relying on donations, staff reported they could only offer a small range of products and could not indulge women's preferences for specific products. The supported accommodation varied on what additional resources they provided, some donating underwear and clothes to women, and others allowed access to washing machines within their services.

Staff reported that they did not discuss menstruation with the women routinely. Those from the supported accommodation sites acknowledged they had little discussion around menstrual health with their female service users, despite sexual health being part of their remit of care.

*"We work in a women's service, . . . for women, about women but I think it's just because of the whole area of it, it's [menstruation] not on our radar its kinda just pushed to one side"*

*(*Staff IDI03 Supported accommodation).

*"The staff need to put out the support side of it for the well-being you know, asking them, are your periods regular, reminding them we've got products, asking them if they need products".* (Staff IDI03 Supported accommodation).

However, whilst they appeared overall to think it would be useful to the women, they felt ill- prepared to have such conversations as they had received no training. Further, they acknowledged that having male workers discuss these issues could be problematic.

*"I think the important thing and why I said I want to talk a little bit about this today is because from a man's perspective it's not as straightforward as it would be for one of my colleagues. I have two women colleagues in the management team, they don't seem to have a problem with it but that's rightfully so, but we need to be aware that maybe 20% of the*

*volunteers are male and for them it's not always straightforward. Erm it's not easy, and . . . I think that's something we need to be aware of and I think us men need to know it's something we need to make sure it's a need that's met and are not convinced we know all the answers to"*

*(*Staff Male IDI01 Foodbank).

**Potential solutions to period poverty issues.** Both women and staff were asked their opinion on potential solutions to reduce period poverty. The most common suggestion, often said unprompted by both women and staff, was that menstrual products should be free. During these conversations, the situation in England was often compared to Scotland where free menstrual products are provided. Frequently, participants suggested that products should be free to all or at least to those on low income via government schemes similar to the provision of free prescriptions for people claiming universal credit, or through a token system.

*P1 "I just think they should hand them out to you wherever you are".*

*P2 "they hand condoms out don't they so what's the difference?"*

*(FGD 02* Supported accommodation)

Improving access to, and increasing the number of, public facilities as well as providing complimentary products were suggestions made or agreed on by many of our participants
Improving menstrual health education was also a frequent recommendation made, with a need for men and boys to be taught as well as girls in school and women in the workplace.

*"lads like grossed out by it urghh and all that . . . . . . . . .so I think that again impacts girls*

*. . . . . .and then it follows them further on into adulthood then the lads are no better educated so they're still growing up with wives and children, 'I don't wanna know I don't wanna know', you know what I mean. So I think education on a whole, start off in schools and maybe in address in science, they touch slightly on the reproductive systems but they don't talk properly about body functions, and I think educate the kids in school first, you know, destigmatise, take all that away"*

(Staff IDI03 Supported accommodation)

A few staff suggested that service providers needed to initiate conversations about menstruation with the women in their care, and they further suggested receiving training in menstrual health to increase their knowledge of products and help signpost women to appropriate medical services.

*"It's having an awareness. Think the staff need to put out the support side of it for the well-being you know, asking them, "are your periods regular?" reminding them we've got products, asking them if they need products, er, you know getting a bit of a feel to whether people are going through the menopause, because that brings other things with it. . . the changes, the hot flushes, stuff that we should be putting in and supporting them around and maybe you know, advising them on it, accessing their GP, talk about HRT or other alternatives. . . and I think just because of the current feel of it all, and it's all kind of behind doors, its forgot about, even with staff now sitting here talking about it, it's all coming out to light its thinking that should be one of our starting blocks"*

(Staff IDI03 Supported accommodation).

## Discussion

Despite the current, largely media driven, push to eradicate period poverty, our study remains one of few evidence-based research studies conducted in a HIC on period poverty within a socioeconomically disadvantaged population. We found that menstruation is viewed negatively and experienced as burdensome by most participants, a situation mirroring that evidenced for girls and women in LMIC [3]. Despite being set in a HIC, the lack of resources clearly impacted on their perceptions and experience of menstruation, adding to the difficulties of achieving good MHH. To complicate matters, the COVID-19 pandemic will continue to make MHH difficult for a growing number of vulnerable women, and organisational support will be increasingly critical for this expanding population for the foreseeable future. The ramifications of these findings are discussed below.

A dominant theme emerging from our study concerned affordability of menstrual products. We found our participants believed better access to products would have a significant influence on their lives. Bobel and Fahs [22] asserted that "It is troubling that activists assert the power of products to impact lives when the research base for these interventions, thus far, is thin" (p975). Our study has provided further evidence indicating the importance of product affordability for vulnerable women which is likely to become even more critical within the UK given the current economic crisis. Although women were able to purchase these items at a relatively lower cost if they visited budget product shops and bought supermarkets' own-brand products, having to purchase menstrual products at any price was a struggle or balancing act at times. This is a recurring theme in other studies conducted in the UK and US [7,8,11] with Barrington and colleagues' systematic review reporting that resource limitations compounded adverse experiences of menstruation [4]. Although menstrual products are essential rather than 'luxury' items, when faced with other spending priorities such as food, lighting and heating they are viewed as more expendable. Menstruators thus must manage their MHH in other ways. Our participants described wearing them for longer than they wished, eking them out by using items such as toilet tissue, or obtaining them by borrowing, stealing or, in one instance, having sex for money in order to purchase pads. Such ways of improvising or obtaining menstrual absorbents are described frequently in the literature from LMIC [23–26] and a few studies conducted in HIC [11–13]. Interestingly, while the systematic review noted numerous studies reflecting on resource limitations, no findings were reported in relation to improvising or overstaying with products [4]. This may result at least in part from our focus on women living in circumstances of deprivation. It is important to highlight that overstaying with menstrual items is associated with discomfort and chafing, also leakage and staining, creating feelings of fear and shame [23,24,27]. There is some evidence from LMIC to suggest that unhygienic materials may increase the risk of genitourinary infections [28–30]. However, the studies in LMIC report that women and girls resort to using rags, mattress stuffing, leaves, grass, ash, etc. [23,31] items which may be less hygienic than those materials described by women in our study which included toilet or tissue paper, socks, cotton wool and sponges. It is not therefore known whether or to what extent improvising or overstaying with these products may increase risk of infection.

Period poverty is also recognised to be a matter of gender inequity [32,33]. Several of our participants drew a parallel of needing sanitary products and requiring condoms yet pointed out that the latter are provided free in many places. Despite sanitary products being made available in schools and also on an ad hoc basis in offices, public houses, shopping centres etc, this was not acknowledged by our participants. We conjecture this may be because the availability is not extensive, nor advertised widely in the circles that impoverished women access, unlike provision of condoms. Indeed, this situation was felt to be unfair, with common

opinion that if men had periods, they would have access to free products as a matter of course. Menstruation is not a choice, and the argument exists that menstrual materials should be regarded as essential items. Furthermore, we speculate that if the men making these decisions in government roles had a better understanding of women's health, such issues may not exist. Although the UK has recently abolished the 'tampon tax', EU countries are still subject to a 5% levy. Similarly, in the US, sales tax paid on the purchase of menstrual products exists because most states consider these products a luxury or, at the least, not a necessity, likely stemming from the historical view that tampons were considered as cosmetics [34].

We had speculated that for menstruators who struggled to purchase sanitary items each month, reusable products might be one solution, hypothesising that, as occurs in many settings in LMIC, the relatively high initial outlay could be borne by donor organisations. We specifically asked women about these products, noting that no individual mentioned them without prior questioning. Although studies suggest that reusable pads and menstrual cups are acceptable and even preferred to other methods in some contexts [35–38], for the most part, our participants were resistant to this idea. This may have been partly due to a lack of information and knowledge which they admitted to, which can be an initial barrier to acceptance [39]. There were also additional challenges to using washable items such as the requirement for additional washing and drying facilities as well as privacy to do this, which our participants using shared accommodation facilities lacked recourse to, and undoubtedly is an obstacle for those living on the streets [40] and for those with heavy bleeding [4]. However, the distaste for seeing and touching menstrual blood voiced by some participants in our study also appeared to be a major hurdle. A recent systematic review and meta-analysis of reusable pads found that participants felt 'disgust' at having to wash menstrual blood in a total of six qualitative studies [41] while Barrington and colleagues review also noted the washing of cloths was 'distasteful' [4]. Further, the drive to use more environmentally friendly products may not be strong amongst vulnerable menstruators in HIC where disposal of commercial sanitary products is relatively easy. Indeed, when questioned, our participants did not see disposal as an issue for them. Many public facilities, including workplaces and entertainment venues, have sanitary bins. Domestically, products can be flushed away in toilets or easily binned in refuse facilities, collected on a regular basis. The environmental effects of disposing of commercial pads or tampons tend to be more visible in LMIC where latrines get blocked, bins are not provided, and materials get discarded on waste or other ground [42,43]. A push to promote reusable products including the menstrual cup, for menstruators on low income may have long term benefits for the individual as well as society. This would require some investment, including information, education and sensitisation, to ensure a successful outcome according to preliminary indications from our small study.

At the time of writing, the UK and most other countries remain in the midst of the Covid-19 pandemic. The pandemic appears to impact on period poverty in a variety of ways as discussed in depth by Crawford and Waldman [17]. Many individuals and families have suffered financially through loss of work or reduced earnings, whether permanently or whilst on furlough. This has increased the proportion of the population who need financial assistance, including to purchase essentials such as menstrual products. Indeed, the economic impact of Covid-19 has been harder on women than men generally, as they earn less, hold less secure jobs and are more likely to be employed in the informal sector. According to the UN policy brief on the impact of COVID on women *"Their capacity to absorb economic shocks is therefore less than that of men"* [44]. Parallel to this, a decrease in supply chain has led to price hikes for many essential household goods, thus also creating more poverty. Repeated lockdowns created severe supply chain bottlenecks that depleted supplies of menstrual products and resulted in price increases, pushing them further out of reach for those on a low income [45]. Anecdotal

evidence from charities dealing with vulnerable individuals and families indicates demand for assistance has risen [46]. One UK charity 'Bloody Good Period' supplies menstrual products to foodbanks, refugees, homeless persons, those fleeing domestic violence, and community support groups amongst others. They report supplying almost six times as many products since the pandemic began [47]. While, to date, no published UK research study has documented the impact of Covid-19 on period poverty, a recent US survey observed that menstrual product insecurity was strongly predicted by pandemic related loss of income, with low income and / or low educational attainment populations at even greater risk [48].

We therefore suggest that organisations offering support to menstruators need to widen access generally due to the increasing proportion of vulnerable populations. They need to provide more menstrual products and make visible the menstrual support they offer. Both women and staff in our study felt this was currently hidden. While there have been recent strides towards combating period poverty in the UK generally, with the abolishment of the 'tampon tax' and free supplies of products to students, this movement needs sustained impetus to achieve greater menstrual equity through supply of free period products or, at the very least, free products to those who can least afford them, following the example of Scotland. Whilst the experience of menstruation will not become a positive one simply through product provision because of the associated physical and psychological burden noted by Barrington et al [4] as a major outcome for women in many studies, at least there will be one less encumbrance for vulnerable women to bear during their menses.

We note some limitations in our study. Unfortunately, due to the Covid-19 pandemic we were unable to achieve the number and range of FGD and IDI that we anticipated. These may have given us additional insight. However, we were struck by the recurring themes that emerged, with remarkably similar viewpoints from women irrespective of data collection method used. This gives us confidence that we have been able to provide a snapshot of the key issues emerging from the perspectives of women living under impoverished circumstances. A second limitation concerns the very differing positionality of participants to the interviewers or moderator. Although some participants may only temporarily be living under difficult circumstances, we were aware of likely variations in socio-economic circumstances to ours both in terms of wealth, occupation, and education status. It must also be remembered that a significant proportion of our FGD participants were ex-offenders who had spent time in prison. We attempted to negate these differences by offering an 'open mind', reminding participants it was their voices we wanted to hear and trying to connect as 'women' who also have experienced (or experience) menstruation. We were particularly mindful of positionality during analysis, adopting a reflexive stance [49]. Further, it is unusual for small qualitative studies to use such a wide range of interviewers, which was necessary due to limitations of time and resources. We attempted to attain some uniformity by devising the tools as a group task, and training together so that we all had similar understanding of the direction to take the conversations. We anticipate that individual traits such as mannerisms and common concerns may have enabled us to connect to individuals on a personal basis facilitating open conversation without changing content radically according to the different interviews.

## Conclusion

Women in our study reported menstruation as a physical, psychological, and financial impediment. The implications from this are that their health and well-being suffer as they balance their MHH needs alongside other day to day hardships. It is likely that the Covid-19 pandemic has increased the number of women who are financially vulnerable. Steps to make menstrual products free for all, or at least for those who suffer financial hardship, are necessary to reduce

the burden of menstruation and help redress gender inequity. Organisations which support impoverished women need to make provision of their menstrual products more visible in order that their clients are aware they can access these products in emergency. While promotion of reusable products such as the menstrual cup may be one part of the solution, this will need to be a longer-term aim with consideration of education and sensitisation to promote attitudinal change. We recommend further research undertaken amongst women in HIC, as MHH and menstruation itself appears to be a pervasive issue, problematic not just amongst women living in LMIC but for women generally.

## Supporting information

**S1 File. Focus group guide.**
(PDF)

**S2 File. In-depth interview guide women.**
(PDF)

**S3 File. In-depth interview guide staff.**
(PDF)

**S4 File. Coding framework.**
(PDF)

## Acknowledgments

We thank the women and staff who participated in this study for their vital contributions. We are extremely grateful to the staff in the support accommodation and foodbanks visited, their engagement and support with this study. Thank you also to Tim Morley for proofreading this article.

## Author Contributions

**Conceptualization:** Diane Exley, Penelope A. Phillips-Howard, Linda Mason.

**Data curation:** Madeleine Boyers, Penelope A. Phillips-Howard, Linda Mason.

**Formal analysis:** Madeleine Boyers, Penelope A. Phillips-Howard, Linda Mason.

**Funding acquisition:** Elizabeth Douglas, Nicola Hawkes, Julie Kelly, Diane Exley, Penelope A. Phillips-Howard.

**Investigation:** Madeleine Boyers, Supriya Garikipati, Alice Biggane, Ciara Kiely, Cheryl Giddings, Penelope A. Phillips-Howard, Linda Mason.

**Methodology:** Madeleine Boyers, Supriya Garikipati, Elizabeth Douglas, Nicola Hawkes, Julie Kelly, Diane Exley, Penelope A. Phillips-Howard, Linda Mason.

**Project administration:** Elizabeth Douglas, Nicola Hawkes, Cheryl Giddings, Julie Kelly.

**Resources:** Elizabeth Douglas, Nicola Hawkes, Julie Kelly, Diane Exley.

**Supervision:** Penelope A. Phillips-Howard, Linda Mason.

**Validation:** Penelope A. Phillips-Howard, Linda Mason.

**Writing – original draft:** Madeleine Boyers, Supriya Garikipati, Penelope A. Phillips-Howard, Linda Mason.

**Writing – review & editing:** Madeleine Boyers, Supriya Garikipati, Alice Biggane, Elizabeth Douglas, Nicola Hawkes, Ciara Kiely, Cheryl Giddings, Julie Kelly, Diane Exley, Penelope A. Phillips-Howard, Linda Mason.

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
