## [Decision Letter · Decision Letter 0]

26 Jan 2022

PONE-D-22-00234"It's not a luxury to prevent having blood run down your legs". The experiences and perceptions of period poverty among impoverished women in an inner-city area of Northwest England.PLOS ONE

Dear Dr. Mason,

Thank you for submitting your manuscript to PLOS ONE. After careful consideration, we feel that it has merit but does not fully meet PLOS ONE’s publication criteria as it currently stands. Therefore, we invite you to submit a revised version of the manuscript that addresses the points raised during the review process. The reviewers have provided some minor comments which nevertheless require careful attention, particulalry on supplying the underlying data.   But I think these should be straightfroward for you to complete.

We look forward to receiving your revised manuscript.

Kind regards,

Alison Parker

Academic Editor

PLOS ONE

Journal Requirements:

"The research was supported by a UKRI ‘Enhancing place-based partnerships in public engagement’ grant."

"The study was funded by  UKRI ‘Enhancing place-based partnerships in public engagement’ grant. The grant holder was Professor P. Phillips-Howard. No grant number was provided by the funder. 

URL https://www.ukri.org/

Reviewers' comments:

Reviewer's Responses to Questions

**Comments to the Author**

1. Is the manuscript technically sound, and do the data support the conclusions?

Reviewer #1: Yes

Reviewer #2: Yes

2. Has the statistical analysis been performed appropriately and rigorously? 

Reviewer #1: N/A

Reviewer #2: N/A

3. Have the authors made all data underlying the findings in their manuscript fully available?

Reviewer #1: No

Reviewer #2: Yes

4. Is the manuscript presented in an intelligible fashion and written in standard English?

Reviewer #1: Yes

Reviewer #2: Yes

5. Review Comments to the Author

Reviewer #1: This is a well-evidenced report of a study investigating the realities of 'period poverty' for women in a city of England. the use of illustrative quotes is sound. It's a shame that the sample size is relatively small due to COVID-19, but I recommend publication as there is so little existing information on these experiences. I do however think that there are a few improvements that could be made to the manuscript in terms of being clearer, and providing some extra details in the Supporting Information; I have highlighted and noted these on a pdf of the manuscript. I don't think these are overly burdensome and I don't think the paper should be accepted until they have been addressed. They are all related to presenting the study more clearly; I don't have any issues with the study itself.

No Supporting Information has been provided as far as I can tell, can these please be made available to reviewers if this paper goes for another round of reviews (I understand this may be the system's, not the authors', fault)? I answered 'No' to Q3 above because I have not seen the data/there is no DOI provided in the manuscript, but the authors do state that "all data are fully available without restriction" and "All relevant data are within the manuscript and its Supporting Information files."

Reviewer #2: I would suggest dramatically reducing the length of your paper's title. It is quite unwieldy. If you insist on using the quote, consider using it as a subtitle, not the primary title.

Also you need at least a paragraph explaining your decision (a perfectly valid one) to refer to "women and girls" instead of "individuals who menstruate," "menstruators," or "women, girls and other individuals who menstruate."

There are some relevant sources the authors have overlooked:

(1) PLAN INTERNATIONAL, PERIODS IN A PANDEMIC: MENSTRUAL HYGIENE MANAGEMENT IN THE TIME OF COVID-19 (2020)

(2) Chris Bobel & Breanne Fahs, From Bloodless Respectability to Radical Menstrual Embodiment: Shifting Menstrual Politics from Private to Public, 45 SIGNS 955 (2020) (for discussion of "menstruators" vs. "women and girls")

(3) Anne Siebert Kuhlman, Unmet Menstrual Hygiene Needs Among Low-Income Women doi: 10.1097/AOG.0000000000003060

(4) Crawford & Waldman, Period Poverty in a Pandemic, https://papers.ssrn.com/sol3/papers.cfm?abstract_id=3692802

(5) Any mention of Scotland's adoption of national legislation providing free menstrual products

You might consider, although this is the weakest of my recommendations, engaging more robustly (in two or three more paragraphs) with the existing literature on period poverty. Some important scholarship has been overlooked.

6. PLOS authors have the option to publish the peer review history of their article (what does this mean?). If published, this will include your full peer review and any attached files.

Reviewer #1: **Yes: **Dani Barrington

Reviewer #2: No

---

## [Author Response · Author response to Decision Letter 0]

15 Apr 2022

Please see response to reviewers letter uploaded:

---

## [Decision Letter · Decision Letter 1]

12 May 2022

PONE-D-22-00234R1Period Poverty: The perceptions and experiences of impoverished women living an inner-city area of Northwest EnglandPLOS ONE

Dear Dr. Mason,

Thank you for submitting your manuscript to PLOS ONE. After careful consideration, we feel that it has merit but does not fully meet PLOS ONE’s publication criteria as it currently stands. Therefore, we invite you to submit a revised version of the manuscript that addresses the points raised during the review process. This manuscript is very close to acceptance now, just a few very minor changes requested by one of the reviewers.

We look forward to receiving your revised manuscript.

Kind regards,

Alison Parker

Academic Editor

PLOS ONE

Journal Requirements:

Reviewers' comments:

Reviewer's Responses to Questions

**Comments to the Author**

1. If the authors have adequately addressed your comments raised in a previous round of review and you feel that this manuscript is now acceptable for publication, you may indicate that here to bypass the “Comments to the Author” section, enter your conflict of interest statement in the “Confidential to Editor” section, and submit your "Accept" recommendation.

Reviewer #1: (No Response)

2. Is the manuscript technically sound, and do the data support the conclusions?

Reviewer #1: Yes

3. Has the statistical analysis been performed appropriately and rigorously? 

Reviewer #1: N/A

4. Have the authors made all data underlying the findings in their manuscript fully available?

Reviewer #1: Yes

5. Is the manuscript presented in an intelligible fashion and written in standard English?

Reviewer #1: Yes

6. Review Comments to the Author

Reviewer #1: Thanks for the opportunity to re-review this paper. I reviewed the version I was sent by email on 10th May, a Word document titled 'Revised Manuscript with tracked changes'. I read this alongside the original version I commented on and noted that not all changes between versions had been tracked. So I'm hoping I did comment on the correct version/the authors just forgot to turn 'Tracked Changes' on for some of the revisions.

Overall, I still think the paper is great, but I have a few small suggestions, which I've annotated (including some typos, I know PLOS One doesn't send proofs to authors so wanted to flag them for you now so they don't make it through to the other end). The only comment on there that I think needs proper 'consideration' (I don't think other stuff is 'controversial', it's mostly grammatical) is around the use of the Bobel and Fahs quote in the Discussion - you suggest your work is at odds with the quote, but I disagree, in fact, your work provides the evidence their quote asks for.

Looking forward to seeing this published!

7. PLOS authors have the option to publish the peer review history of their article (what does this mean?). If published, this will include your full peer review and any attached files.

Reviewer #1: **Yes: **Dani Barrington

---

## [Author Response · Author response to Decision Letter 1]

17 May 2022

Dear Alison Parker

Thank you very much for reviewing our paper, ‘Period Poverty: The perceptions and experiences of impoverished women living an inner-city area of Northwest England’ again. 

We wish to think Reviewer 1 for taking the time to provide such meticulous feedback. It has been very helpful indeed and hopefully we have done justice to her efforts to improve our paper.

All grammatical suggestions have been addressed (we have not itemised in this response), and having been alerted to errors we have, in addition, incorporated suggestions from a proof-reader. There are two issues we provide detail on. 

1) The suggestion to group the US studies versus other studies was attempted on an early draft for the very reasons outlined by Reviewer 1. However, in doing this the narrative became quite complicated. We reverted to a narrative that groups studies by method and findings, which does require mention of the geography of each study, but overall reads, in our opinion, more straightforwardly.

2) We understand the point made regarding our original response to the Bobel and Fahs article. On re-reading we now agree that Bobel and Fahs were arguing the need for more evidence before assertions are made. Our study helps in providing additional insight to the topic and we now have clarified this point. “A dominant theme emerging from our study concerned affordability of menstrual products. We found our participants believed better access to products would have a significant influence on their lives. Bobel and Fahs [22] asserted that “It is troubling that activists assert the power of products to impact lives when the research base for these interventions, thus far, is thin” (p975). Our study has provided further evidence indicating the importance of product affordability for vulnerable women, which is likely to become even more critical within the UK given the current economic crisis”

We apologise for the loss of some track changes in the previous version sent which we appreciate must have made this more difficult and time consuming to check our amendments. Unfortunately, we have been unable to highlight this last set of track changes in a different format, which would have made it easier to check and we thank you for your patience.

We hope this response meets with your and the reviewers approval.

Best wishes,

Linda Mason

Senior Research Fellow

Liverpool School of Tropical Medicine, Pembroke Place, L3 5QA, Liverpool,

---

## [Editor Report · Decision Letter 2]

19 May 2022

Period Poverty: The perceptions and experiences of impoverished women living an inner-city area of Northwest England

PONE-D-22-00234R2

Dear Dr. Mason,

We’re pleased to inform you that your manuscript has been judged scientifically suitable for publication and will be formally accepted for publication once it meets all outstanding technical requirements.

Kind regards,

Alison Parker

Academic Editor

PLOS ONE
---

## [Editor Report · Acceptance letter]

16 Jun 2022

PONE-D-22-00234R2 

Period Poverty: The perceptions and experiences of impoverished women living in an inner-city area of Northwest England 

Dear Dr. Mason:

I'm pleased to inform you that your manuscript has been deemed suitable for publication in PLOS ONE. Congratulations! Your manuscript is now with our production department. 

Kind regards, 

on behalf of

Dr. Alison Parker 

Academic Editor

PLOS ONE